# CSF1R blockade induces macrophage ablation and results in mouse choroidal vascular atrophy and RPE disorganization

Xiao Yang[1], Lian Zhao[1], Maria M Campos[2], Mones Abu-Asab[2], Davide Ortolan[3], Nathan Hotaling[3], Kapil Bharti[3], Wai T Wong[1]*

[1]Section on Neuron-Glia Interactions in Retinal Disease, National Eye Institute, National Institutes of Health, Bethesda, United States; [2]Section on Histopathology, National Eye Institute, National Institutes of Health, Bethesda, United States; [3]Section on Ocular and Stem Cell Translational Research, National Eye Institute, National Institutes of Health, Bethesda, United States

**Abstract** The choroid, which provides vascular supply to the outer retina, demonstrates progressive degeneration in aging and age-related macular degeneration (AMD). However mechanisms that maintain or compromise choroidal homeostasis are obscure. We discovered that the ablation of choroidal macrophages via CSF1R blockade was associated with choroidal vascular atrophy and retinal pigment epithelial (RPE) changes including structural disruption, downregulation of visual cycle genes, and altered angiogenic factor expression. Suspending CSF1R blockade following ablation enabled spontaneous macrophage regeneration, which fully restored original macrophage distributions and morphologies. Macrophage regeneration was accompanied by arrested vascular degeneration and ameliorated pathological RPE alterations. These findings suggest that choroidal macrophages play a previously unappreciated trophic role in maintaining choroidal vasculature and RPE cells, implicating insufficiency in choroidal macrophage function as a factor in aging- and AMD-associated pathology. Modulating macrophage function may constitute a strategy for the therapeutic preservation of the choroid and RPE in age-related retinal disorders.

*For correspondence: wongw@alum.mit.edu

**Competing interests:** The authors declare that no competing interests exist.

## Introduction

The choroid of the mammalian eye comprises a dense vascular network located external to the retina and provides necessary material transport to the outer retina across the retinal pigment epithelial (RPE) layer constituting the outer blood retinal barrier (*Fields et al., 2019*). The functional significance of the choroidal supply is underscored by the implication of choroidal vascular atrophy as a significant contributor to degenerative diseases of the outer retina, particularly age-related macular degeneration (AMD) (*Arya et al., 2018*; *Sohn et al., 2019*), a current leading cause of blindness in the developed world (*Bourne et al., 2018*). While progressive loss of choroidal vasculature has been observed with aging and AMD (*Keenan et al., 2020*; *Spaide, 2009*), the mechanisms that either actively drive choroidal atrophy or fail to sustain choroidal homeostasis are not well understood, and treatments that can successfully preserve choroidal structure are unavailable.

Like in vascular networks located at border regions surrounding the brain within the dura mater, subdural meninges and choroid plexus (*McMenamin, 1999b*; *Van Hove et al., 2019*), the choroid of the eye contains a large constituent population of innate immune cells in the form of resident choroidal macrophages (*Forrester et al., 2010*; *McMenamin, 1999a*). Under normal conditions, these macrophages are distributed across the entire choroid in the perivascular space in close juxtaposition with choroidal vessels. As they possess ramified cellular processes that demonstrate dynamic behavior, choroidal macrophages can potentially survey the perivascular environment and exchange

signals with nearby vascular and RPE cells (*Kumar et al., 2014*). While activated choroidal macrophages have been implicated in contributing to pathological changes in posterior uveitis (*Jiang et al., 1999*) and choroidal neovascularization (*Cherepanoff et al., 2010*), their endogenous functions under healthy conditions are obscure, and if and how they may contribute to homeostasis in the choroid are not understood.

Dysfunction in choroidal macrophages has been linked to choroidal vascular alterations detected in aging and in AMD. Altered macrophage morphology and distribution have been correlated with attenuation of the choriocapillaris in the aged mouse choroid (*Kumar et al., 2014*), and increased density and expression of activation markers in macrophages have been documented with the onset of AMD in human eyes (*McLeod et al., 2016*), suggesting a causal relationship between macrophage changes and vascular loss. The implication of choroidal vascular loss as a driver of drusen accumulation and RPE disruption in AMD pathogenesis (*Biesemeier et al., 2014*; *Lengyel et al., 2004*; *Mullins et al., 2011*) have prompted strategies aimed at maintaining choroidal vascular health and improving vascular supply to the outer retina (*Coleman et al., 2018*; *Whitmore et al., 2015*), but the intrinsic mechanisms that maintain vascular and RPE integrity in the choroid have not been elucidated, limiting translational efforts.

We investigated here the homeostatic relationship that choroidal macrophages have with choroidal vasculature and RPE cells. We found that pharmacological CSF1R inhibition, which has been found to ablate tissue-resident innate immune cells in different CNS compartments, but relatively sparing circulating monocytes, induced the ablation of choroidal macrophages in a sustained manner. We discovered using in vivo structural imaging and histopathological and molecular analyses that choroidal macrophage ablation was associated with progressive vascular atrophy and with multiple structural and functional changes in the RPE layer, in ways resembling those observed in the aging and AMD eyes. We also discovered that residual choroidal macrophages, following depletion, can undergo regeneration upon suspension of CSF1R inhibition. Although normally non-proliferative in nature, residual macrophages following depletion proliferated in situ and underwent morphological maturation to fully reconstitute features of the endogenous macrophage population. These regenerated macrophages appear to restore endogenous macrophage homeostatic functions, as their repopulation of the choroid was associated with the arrest of vascular atrophy and the amelioration of pathological RPE alterations.

Taken together, our findings implicate a previously unappreciated constitutive function for resident choroidal macrophages in maintaining structural and functional homeostasis of choroidal vasculature and RPE cells and the ability of macrophages to regenerate the full complement of cells following depletion. These suggest a potential etiological mechanism of macrophage insufficiency for pathological choroidal vascular atrophy and highlight choroidal macrophage renewal as a phenomenon that may be potentially harnessed in regenerative strategies in the outer retina. Modulating choroidal macrophage presence and phenotype may be therapeutically effective in maintaining choroidal vasculature and consequently delaying AMD progression.

## Results

### Age-related progressive choroidal thinning and vascular atrophy are associated with alterations in choroidal macrophages

Aging has previously been associated with negative changes in the human choroid, including reductions in choroidal thickness on in vivo imaging (*Entezari et al., 2018*; *Gattoussi et al., 2019*; *Wakatsuki et al., 2015*) and decreased choriocapillaris density on histopathological analysis (*Ramrattan et al., 1994*), however mechanisms underlying this association are largely unelucidated. To examine whether similar age-related choroidal changes were present in our model of the mouse choroid, we examined and compared the vascular structure of the young adult (3 month-old) with aged (24 month-old) BALB/cJ mice. Consistent with observations in human subjects, the choroid of aged mice showed significantly decreased thickness on in vivo enhanced-depth OCT imaging in both the nasal and temporal quadrants relative to young adult mice (*Figure 1A,B*). Quantitative analysis of choriocapillaris structure in sclerochoroidal flatmounts of DiI-perfused animals also showed decreased vascular area coverage (*Figure 1C,D*). We observed that these age-related changes in choroidal vasculature were associated with alterations in macrophages located in the

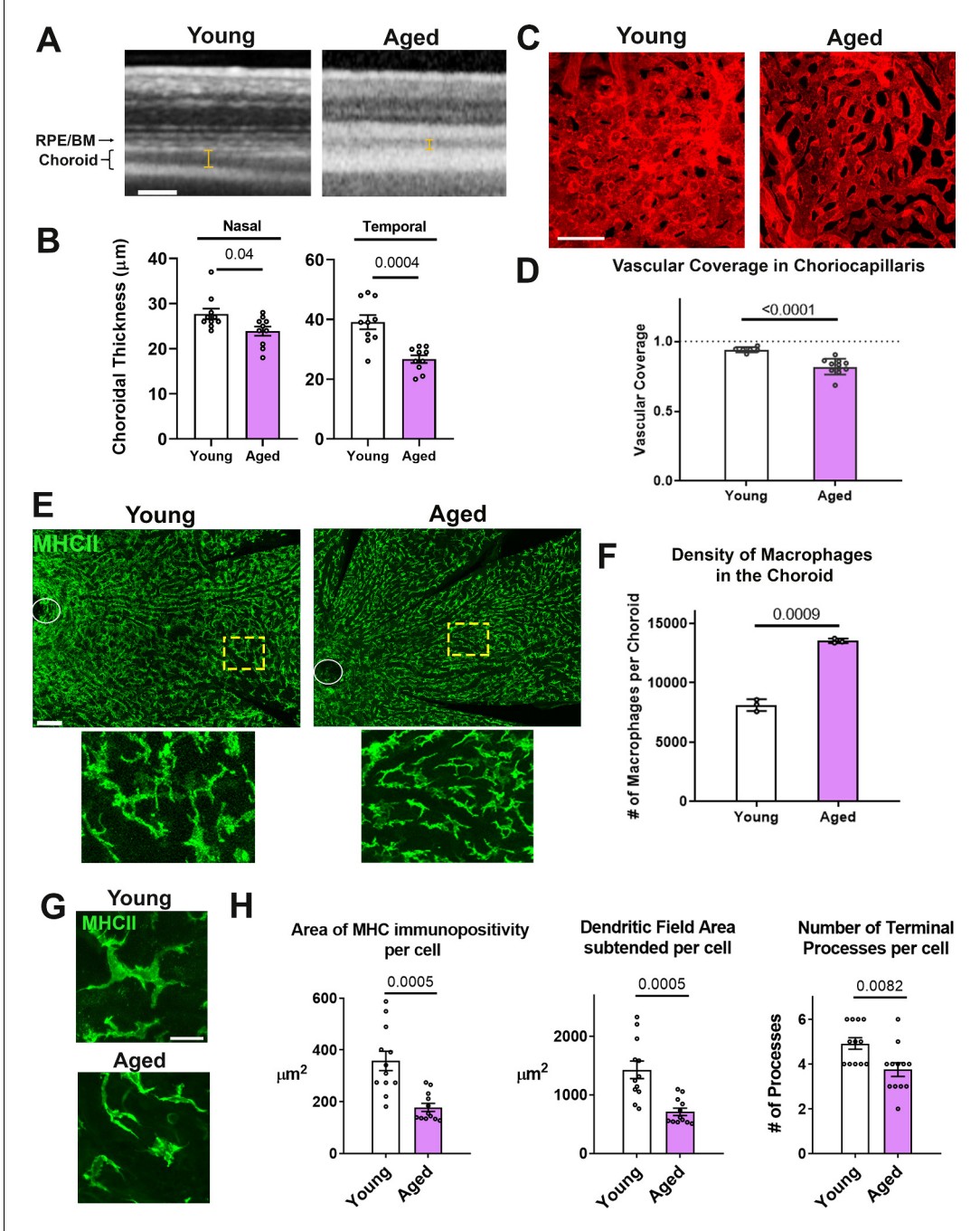

**Figure 1.** Age-related progressive choroidal thinning and vascular atrophy are associated with alterations in choroidal macrophages. (**A**) Young adult (3 month old) and aged (24 month-old) wild type BALB/cJ mice were imaged with enhanced depth imaging optical coherence tomography (EDI OCT) imaging; measurements made of choroidal thickness (indicated by I-bars). Scale bar = 100 µm. (**B**) Choroidal thickness measurements, assessed in the temporal and nasal quadrants (300 µm temporal and nasal respectively from the optic nerve on the horizontal meridian), were significantly decreased in the aged group. (**C**) Choriocapillaris vascular structure was analyzed in sclerochoroidal flatmounts following systemic DiI perfusion. Scale bar = 50 µm. (**D**) The fractional area coverage by choriocapillaris vessels was significantly reduced in the aged animals. (**E**) Choroidal macrophages were visualized by immunohistochemical staining for MHCII in sclerochoroidal flatmounts. Scale bar = 100 µm. Panels show representative images with insets showing high-magnification views. Ellipses mark the position of the optic nerve. (**F**) Macrophage density was markedly increased in the aged group. n = 3 animals per group. (**G**) High magnification images of choroidal macrophages showing reduced ramification in aged vs. young adult animals. Scale bar = 20 µm. (**H**) Quantitative morphological analyses of macrophages from aged animals showed decreased mean cell size (defined as the area of MHCII-immunopositivity per cell), cell coverage (dendritic field area subtended by a single cell), and mean process number (number of terminal processes per cell). P values indicate comparisons relative to control with an unpaired t-test with Welch's correction, n = 4 animals per group.

perivascular space of the vascular choroid. Examination of MHCII-immunopositive macrophages in sclerochoroidal flatmounts showed that these were significantly increased in overall density in the aged choroid (*Figure 1E,F*) compared with the young choroid, but the macrophages in the aged choroid demonstrated decreased ramification as evidenced by significantly decreased cell area, dendritic field area, and number of cellular processes per cell (*Figure 1G,H*). This association suggested that age-related changes in choroidal macrophages may be contributory to negative alterations of the vascular choroid.

## Sustained depletion of resident macrophages in the choroid is accompanied by progressive choroidal thinning and vascular atrophy

To investigate the endogenous contribution of resident macrophages to the structure of the adult mouse choroid, 3 month old young adult mice were administered a diet containing PLX5622, an inhibitor of the macrophage-expressed receptor CSF1R, which regulates macrophage survival (*MacDonald et al., 2010*). Compared with the choroid of control animals fed a regular diet, which had a typical network of ramified MHCII+ and IBA1+ resident macrophages (*Kumar et al., 2014*; *McMenamin, 1999a*), the choroid in PLX5622-administered animals demonstrated a widespread depletion of macrophages in all topographical areas, and possessed only a small number of residual MHCII+, IBA1+ cells showing a non-ramified morphology (*Figure 2A*). Following 1 week of PLX5622 administration, the density of MHCII+, IBA1+ macrophage decreased to ≈10% of that in control animals and was sustained at these levels with continuous PLX5622 administration for up to 7 weeks (*Figure 2B*). Expression of *Iba1* mRNA in sclerochoroidal tissue was correspondingly decreased from baseline levels at week 1 and week 7 of PLX5622 administration (*Figure 2C*), indicating that CSF1R inhibition resulted in a rapid and comprehensive depletion of resident myeloid cells from the choroid.

As macrophage depletion has been causally associated with vascular changes in pathological contexts in the retina (*Sakurai et al., 2003*) and elsewhere (*Jarosz-Biej et al., 2018*), but is of lesser consequence to normal adult retinal vasculature (*Wang et al., 2016*), we investigated whether choroidal macrophages exert a trophic influence in the healthy adult vascular choroid by examining the effects of macrophage depletion. In vivo enhanced-depth OCT images captured in a longitudinal manner before and during PLX5622 administration revealed significant and progressive reductions in choroidal thickness across the entire area of the choroid; measurements made in both the nasal and temporal quadrants showed time-dependent decreases with sustained PLX5622 administration, declining to ≈70% of baseline values by 7 weeks (*Figure 3A,B*). By contrast, the structure of the laminated retina and total retinal thickness measurements in the same experimental animals were unchanged during PLX5622 administration (*Figure 3C*).

Histological analyses of choroidal vascular structure in stained retinochoroidal methacrylate-embedded sections demonstrated corresponding significant decreases in the mean thickness of the vascular layer of the choroid (*Figure 3D,E*). Quantitative analysis of vascular lumina found reductions in the mean cross-sectional area of vascular lumina and the mean density of choroidal vessels (*Figure 3F*). To visualize the choriocapillaris, vessels were perfused and labeled with DiI lipophilic dye. Quantitative analysis of choriocapillaris structure in sclerochoroidal flatmounts of DiI-perfused animals showed decreased vascular area coverage by choriocapillaris vessels (*Figure 3G,H*). Together these findings demonstrate that sustained depletion of choroidal macrophages was associated with generalized vascular atrophy in the adult mouse choroid, a feature not observed in the adult retina. These vascular changes were not associated with a marked loss and disorganization of the population of smooth muscle actin (SMA)-expressing choroidal perivascular cells in the choroid (*Condren et al., 2013*), as evidenced by their maintained organization and morphology in macrophage-depleted animals (*Figure 3—figure supplement 1A,B*).

## Sustained depletion of choroidal macrophages is associated with aberrant RPE structure, gene expression, and function

As choroidal macrophages have been implicated in influencing RPE physiology in the context of macrophage-RPE-vascular interactions (*Shi et al., 2011*; *Wang et al., 2015*), we investigated whether macrophage depletion from the healthy adult choroid exerted influences on the overlying RPE layer. Following 7 weeks of PLX5622 administration, immunohistochemical analysis of the RPE

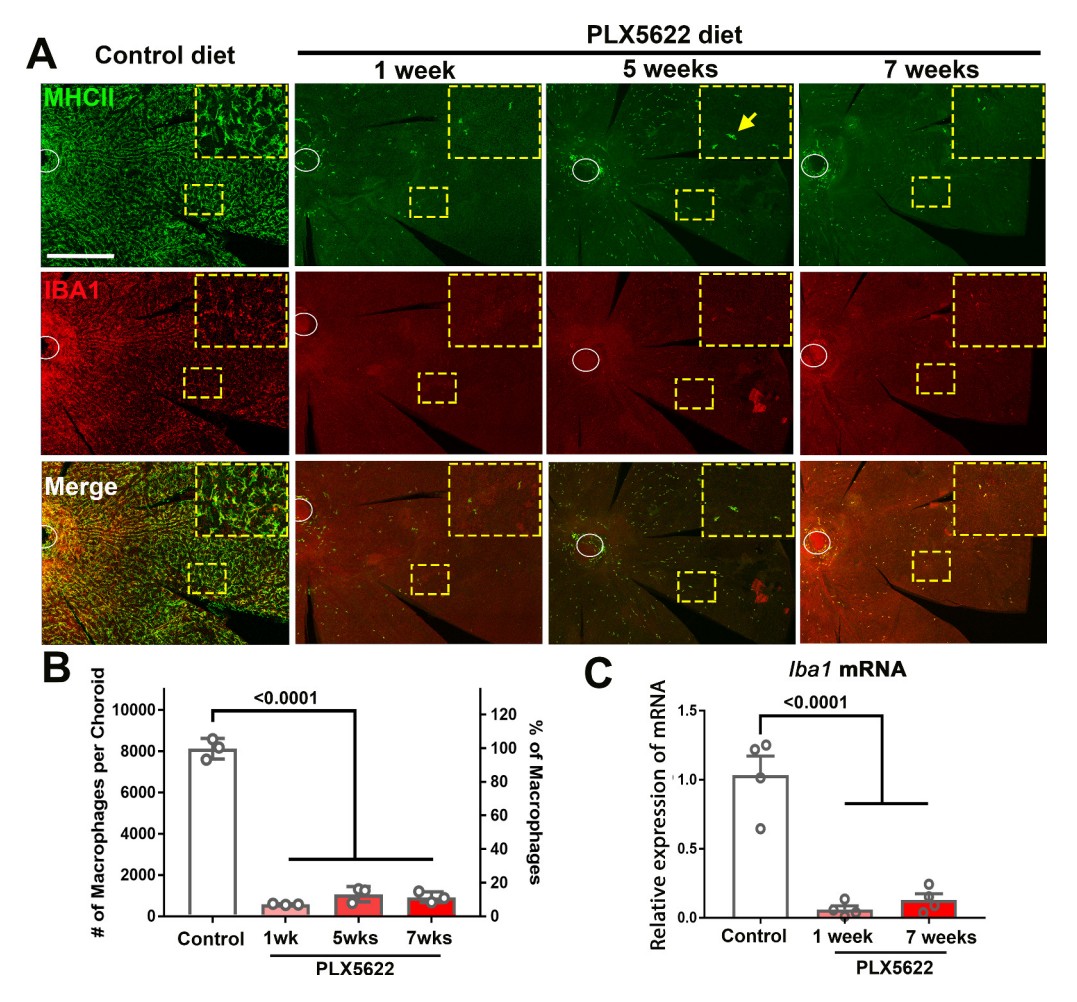

**Figure 2.** CSF1R inhibition via PLX5622 administration induces a rapid and sustained depletion of resident macrophages in the adult mouse choroid. (A) Three-month old wild type Balb/cJ mice were administered diet containing PLX5622 (at 1200 parts per million) continuously for up to 7 weeks. Controls consisted of age-matched animals maintained on standard mouse chow. Resident macrophages in the choroid were visualized by immunohistochemical staining for MHCII (*green*) and IBA1 (*red*) in sclerochoroidal flatmounts. Panels show representative images with insets showing high-magnification views. Ellipses mark the position of the optic nerve. The choroid of control animals demonstrated a dense network of IBA1+, MHCII + resident macrophages with ramified, dendritiform morphology. Macrophage numbers were markedly decreased following 1 week of PLX5622 administration which remained reduced at similar levels for up to 7 weeks of administration. The few residual macrophages showed reduced ramification. Scale bar = 500 μm. (B) Quantitative counts of total MHC-II$^+$ macrophages across the entire choroid revealed a ≈90% reduction in macrophage numbers by 1 week of PLX5622 treatment, which was sustained with continued treatment. (C) Quantitative PCR analyses of the whole RPE-choroid complex demonstrated a corresponding reduction in the expression of *Iba1* mRNA, corresponding to the depletion of Iba1-expressing macrophages. P values indicate comparisons relative to control, 1-way ANOVA, n = 3 animals per treatment group.

layer showed decreased immunopositivity of RPE65, a constitutively expressed, RPE-specific enzyme that is required to maintain visual cycle function (*Redmond et al., 1998*). Immunopositivity for RPE65, which was prominent and uniform in control animals, was decreased in depleted animals in a patchy, heterogenous manner, with a reduction in overall expression (*Figure 4A,B*). These changes corresponded to a decrease in *Rpe65* mRNA expression (*Figure 4—figure supplement 1A*), as well as in significantly decreased mRNA expression of genes mediating visual cycle function (*Lrat, Rlbp1*), RPE differentiation (*Mitf*), and tight junction structure (*Tjp1/Zo1*) (*Figure 4—figure supplement 1A–C*). Electron microscopic analysis of RPE cell structure showed disrupted apical microvilli structure and increased vacuolation in depleted animals relative to control animals (*Figure 4C*). In depleted animals, increased irregularity in the cell areas of individual RPE cells was noted on F-actin labeling; the mean overall RPE cell density was significantly decreased, corresponding to an increased

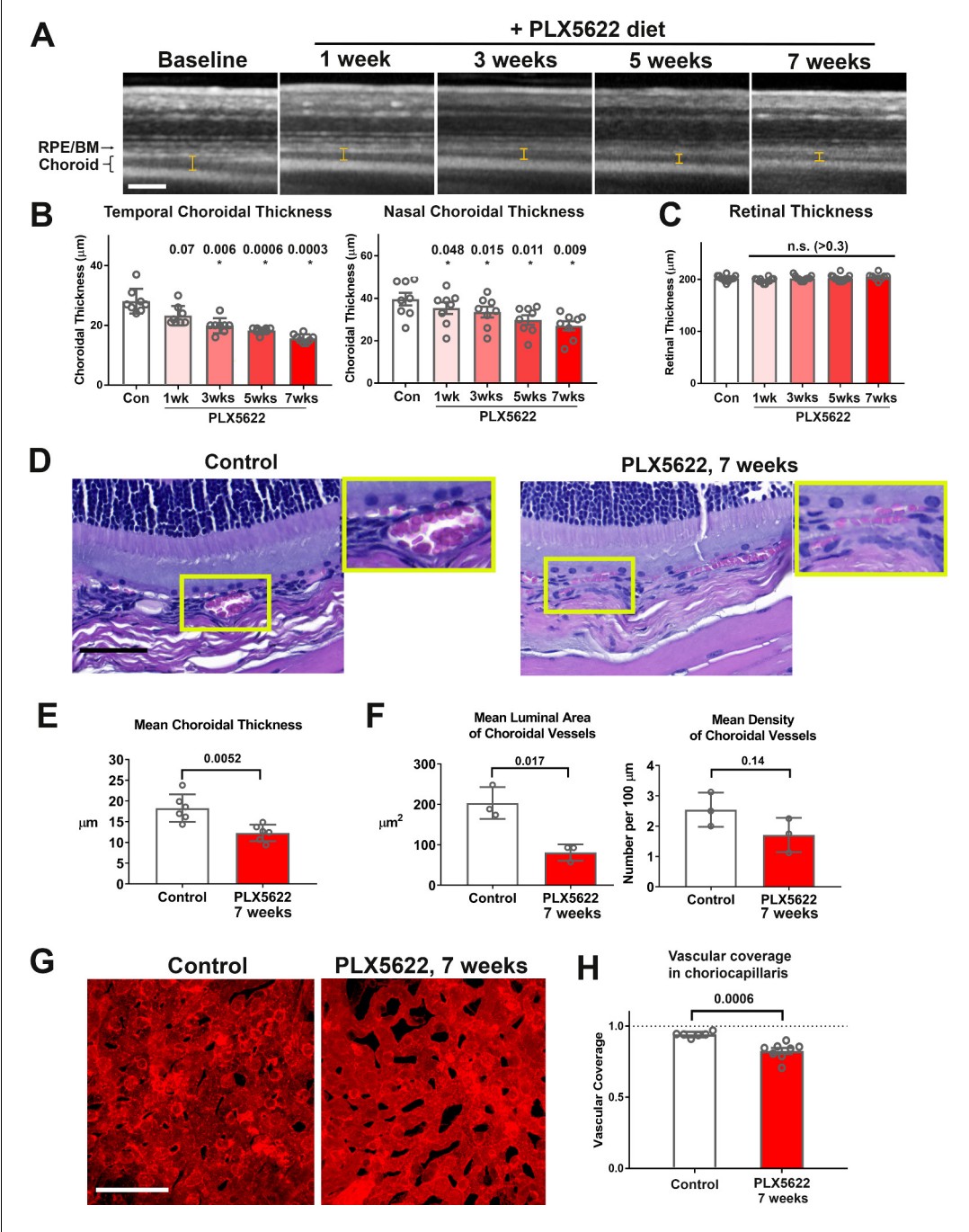

**Figure 3.** Long-term depletion of resident choroidal macrophages is accompanied by progressive choroidal thinning and vascular atrophy. (A) Animals administered PLX5622-containing diet continuously over 7 weeks were followed with longitudinal in vivo enhanced depth imaging optical coherence tomography (EDI OCT) imaging; I-bars show the measurements made of choroidal thickness. Scale bar = 100 μm. (B) Choroidal thicknesses measured in the temporal and nasal quadrants (at 300 μm temporal and nasal respectively from the optic nerve at the horizontal meridian) demonstrated progressive and significant decreases from baseline. (C) Total retinal thickness (measured from the RPE layer to the vitreal surface at the internal limiting membrane) were stable over the same period. P values in B, C were computed for comparisons relative to baseline using 1-way ANOVA test, n = 10 eyes from five animals at each time-points. (D–F) Comparative histological analysis of the choroid in control and depleted animals showed decreases in overall choroidal thickness and vascular density. Panels show representative images with insets (*yellow boxes*) showing high-magnification views. Scale bar = 50 μm. Mean total choroidal thickness was significantly reduced with macrophage depletion (E), as was the mean cross-sectional area of vascular lumina (F), with a trend towards decreased density of choroidal vessels. n = 6 eyes from three animals per treatment group. (G, H) Choriocapillaris vascular structure was imaged in flat-mounted sclerochoroidal tissue following vascular labeling with DiI perfusion; the fractional area coverage by

*Figure 3 continued on next page*

Figure 3 continued

vessels in the choriocapillaris was significantly reduced following 7 weeks of macrophage depletion. Scale bar = 50 μm. n = 8 eyes from four animals per treatment group. P values in E), (F), and H) were computed with an unpaired t-test with Welch's correction.

The online version of this article includes the following figure supplement(s) for figure 3:

**Figure supplement 1.** Long-term depletion of resident choroidal macrophages does not result in marked depletion or disorganization of smooth muscle actin (SMA)-expressing perivascular smooth muscle cells or choriocapillaris pericytes.

proportion of enlarged cells (cell area >900 μm$^2$) (*Figure 4D*). Enlarged cells in depleted animals showed an abnormal increase in cellular nuclei (usually 3–4 nuclei), a feature not observed in control animals (*Figure 4E*). While the proportions of mono- and binucleated cells which constituted the majority of RPE cells were unchanged with depletion, RPE cells demonstrated a generally increased intracellular vacuolation on light microscopy. These changes in RPE morphology and expression of genes mediating RPE function were associated with concurrent reductions in electroretinographic responses in the form of decreased scotopic a- and b-wave amplitudes (*Figure 4F*). Together, these data indicate that resident choroidal macrophages constitutively maintain aspects of RPE organization and gene expression that are important for RPE function in the healthy adult animal.

## Altered angiogenic factor expression in the RPE-choroid complex is associated with choroidal macrophage depletion

As vascular structure of the adult choroid requires constitutive trophic support provided by angiogenic factors, in particular vascular endothelial growth factor (VEGF), for homeostatic maintenance (*Kurihara et al., 2012*; *Saint-Geniez et al., 2009*), we investigated whether choroidal macrophage depletion influences endogenous angiogenic factor expression in the RPE-choroid complex. We found that mRNA levels of *Vegfa* and *Vegfc* (*Figure 5A*) and protein levels of VEGF (*Figure 5B*) were significantly reduced in the RPE-choroid complex following 7 weeks of macrophage depletion. Consistent with this, we observed that immunopositivity for VEGF in cross-sections of the RPE-choroid complex, which is detected predominantly in the RPE layer in control animals, was decreased following macrophage depletion (*Figure 5C*). We also analyzed the expression of pigment epithelium-derived factor (PEDF), a secreted, multi-functional anti-angiogenic agent capable of suppressing choroidal neovascularization (*Amaral and Becerra, 2010*; *Mori et al., 2001*). We found that *Serpinf1* (*Pedf*) mRNA and PEDF protein levels (*Figure 5D,E*), as well as PEDF immunopositivity in the RPE layer, were all significantly increased following macrophage depletion (*Figure 5F*). In contrast, mRNA and protein expression levels of platelet-derived growth factor (PDGF) ligands, PDGF-AA and PDGF-BB, factors implicated in pericyte recruitment and vascular stabilization in the retina (*Lindahl et al., 1997*), were not markedly altered by macrophage depletion (*Figure 5G,H*). These data suggest that altered expression of trophic factors generally, and a decrease in the ratio of VEGF/PEDF secretion specifically (*Pollina et al., 2008*), occurs in the RPE-choroid with macrophage depletion, and may constitute a mechanism contributing to the choroidal vascular atrophy observed.

Past studies had indicated that RPE-derived VEGF are required for vascular survival in the choroid (*Marneros et al., 2005*; *Saint-Geniez et al., 2009*), and that TGF-β1 can induce VEGF expression in the RPE cells (*Nagineni et al., 2003*). We found that *Tgfb1* mRNA levels and TGFB1 immunopositivity were reduced in the choroid following 7 weeks of macrophage depletion (*Figure 5—figure supplement 1A,B*). Taken together, these data suggested that changes in angiogenic factor expression may underlie macrophage-mediated vascular support.

## Macrophages in the choroid demonstrate spontaneous regeneration to repopulate the choroid following depletion

We and others have previously demonstrated that microglia normally resident in the adult retina can demonstrate spontaneous repopulation following depletion (*Huang et al., 2018*; *Zhang et al., 2018*). However, whether choroidal macrophages in the adult choroid can regenerate is unknown. To investigate this, we depleted choroidal macrophages in young adult mice using a PLX5622-containing diet for 3 weeks, a time-point when significant choroidal thinning has occurred (depletion phase), and then reverted these animals to a standard diet for another 4 weeks (repopulation phase) (*Figure 6A*). An age-matched group fed a standard diet served as a control. We observed that when

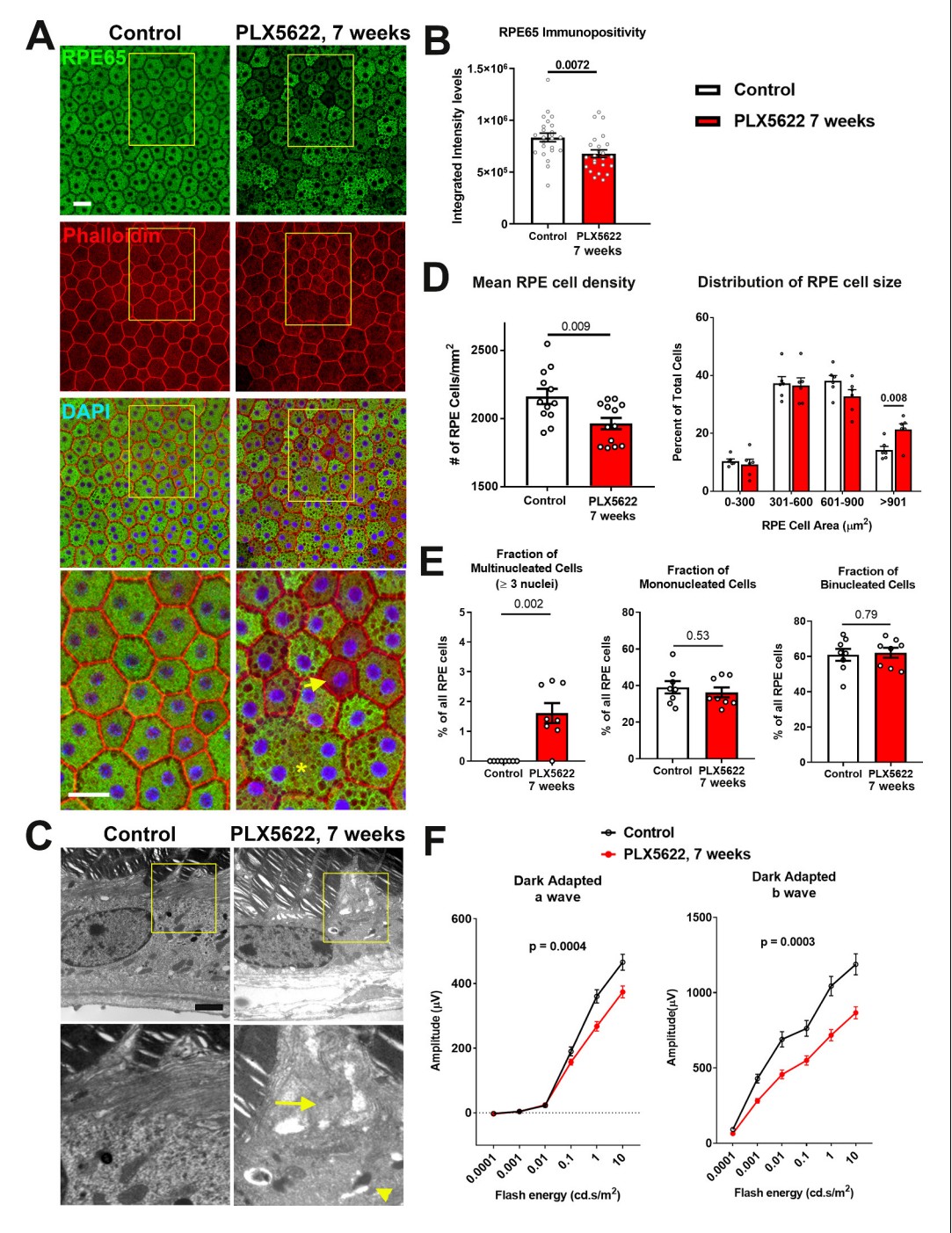

**Figure 4.** Long-term depletion of choroidal macrophages is associated with alterations in RPE structure and function. (A) Immunohistochemical analyses of the RPE monolayer in sclerochoroidal flatmounts in control and PLX5622-treated animals (for 7 weeks) were performed with RPE65 (green), conjugated phalloidin to stain F-actin (red), and DAPI (blue). High-magnification views (yellow box inset) highlight large multinucleated (>3 nuclei) cells (asterisk *), local reductions of RPE65 immunopositivity (arrow), and generally increased intracellular vacuolation. Scale bar = 20 μm. (B) Quantitative analysis of RPE65 immunopositivity in RPE. (C) Transmission electron microscopic analysis of RPE cells show in PLX5622-treated animals disrupted RPE apical microvilli structure (arrow) and increased intracellular vacuolation (arrowhead). Quantitative comparison of RPE morphological features from control and depleted animals of (D) RPE cell density, mean and distribution, and (E) the proportion of multinucleated (≥3 nuclei), mononucleated and binucleated RPE cells. P values in B, D, E correspond to comparisons made using an unpaired t-test with Welch's correction. n = 4 to 6 animals in each treatment group. (F) Comparison of scotopic electroretinographic responses between control and depleted animals showed significantly decreased a- and b-wave amplitudes in the PLX5622-treated group. P values correspond to comparisons made using a 2-way ANOVA test, n = 10 to 16 eyes from 5 to 8 animals in each treatment group.

*Figure 4 continued on next page*

*Figure 4 continued*

The online version of this article includes the following figure supplement(s) for figure 4:

**Figure supplement 1.** mRNA expression levels of genes related to RPE function are altered following transient and sustained depletion of choroidal macrophages.

macrophage-depleted animals were returned to a standard diet, MHCII+, IBA1+ macrophages reappeared rapidly in the choroid, with cell density reaching 50% of control levels after 1 week of repopulation, attaining 100% of control levels at 4 weeks, and remaining stable at that level thereafter (*Figure 6B,C*). This increase in macrophage cell numbers was at least in part enabled by local proliferation of residual IBA1+ cells; while resident macrophages in the choroid of control animals were mostly non-proliferative and negative for the proliferation marker Ki67, prominent numbers of Ki67 +,IBA1+ cells were detected at 1 week of repopulation when macrophage numbers were rising sharply (*Figure 6—figure supplement 1A,B*). At 4 weeks repopulation, Ki67+ cell counts decreased back to control levels at a time when macrophage numbers have recovered to control levels. We cannot rule out that infiltrating monocytes from the circulation may contribute to the repopulation process, but it is likely that repopulated macrophages are, at least in part, originating from the proliferation of the local residual macrophages. These observations reveal a previously uncharacterized regenerative capability of choroidal macrophages in which local proliferation of myeloid cells in the choroid likely contributes to the recovery of the macrophage population following depletion.

During the repopulation process, choroidal macrophages demonstrated features of phenotypic maturation in situ. Although all choroidal macrophages in the control animals showed immunopositivity for both MHCII and IBA1, we observed that a fraction of IBA1+ macrophages at 1 week in the repopulation phase were MHCII-negative, while no MHCII+ macrophages were IBA1-negative (*Figure 6D*). This fraction of MHCII-, IBA1+ macrophages diminished with time and was undetectable at 6 weeks of repopulation, indicating MHCII as a later marker acquired by repopulating macrophage during maturation in situ. Repopulating macrophages also demonstrate maturation in terms of their morphological features. At 1 week of repopulation, although IBA1+ cell numbers had risen substantially to about 50% of control levels, most macrophages showed a deramified morphology with only a few short rudimentary processes. From 2 to 6 weeks of repopulation, macrophages developed progressively increased ramification, increasing in cell area, number of cellular processes, and dendritic field area, to acquire morphological features that were similar from choroidal macrophages in control animals (*Figure 6—figure supplement 1C–F*).

## Recovery of macrophages via regeneration is accompanied by an arrest in depletion-related degenerative changes in the RPE-choroid

To investigate the functional implications of choroidal macrophage regeneration following depletion, we evaluated the structure of the RPE-choroid complex during and following repopulation. We compared (1) animals that were subjected to macrophage depletion for 3 weeks and then allowed to repopulate for another 4 weeks (the depletion/repopulation group) with (2) animals subjected to continuous depletion for a total of 7 weeks (the continuous depletion group) (*Figure 7A*). We found on OCT imaging that while choroidal thicknesses in the continuous depletion group declined progressively over 7 weeks, those in the depletion-repopulation group stabilized upon the initiation of macrophage repopulation, and were significantly higher than those in the continuous depletion group at the end of the 7 week experiment (*Figure 7B*). Assessment of choriocapillaris structure in choroidal flatmounts also found that areal coverage by choriocapillaris vessels decreased progressively over 7 weeks in the continuous depletion group, those in the depletion-repopulation group stabilized between 3 to 7 weeks during macrophage repopulation (*Figure 7C*). These findings suggested that repopulating macrophages functionally support the vascular structure of the choroid, further implicating the constitutive contribution of choroidal macrophages to vascular maintenance.

We similarly assessed the function of repopulating macrophages in the support of RPE layer in these two experimental groups (*Figure 8A*). In the continuous depletion group, mean RPE density at 3 weeks of depletion were similar to untreated controls but decreased at 7 weeks of depletion; in the depletion-repopulation group, mean RPE density was largely unchanged across 7 weeks (*Figure 8B*). Analysis of the distribution of individual RPE cell areas showed that while RPE cells in

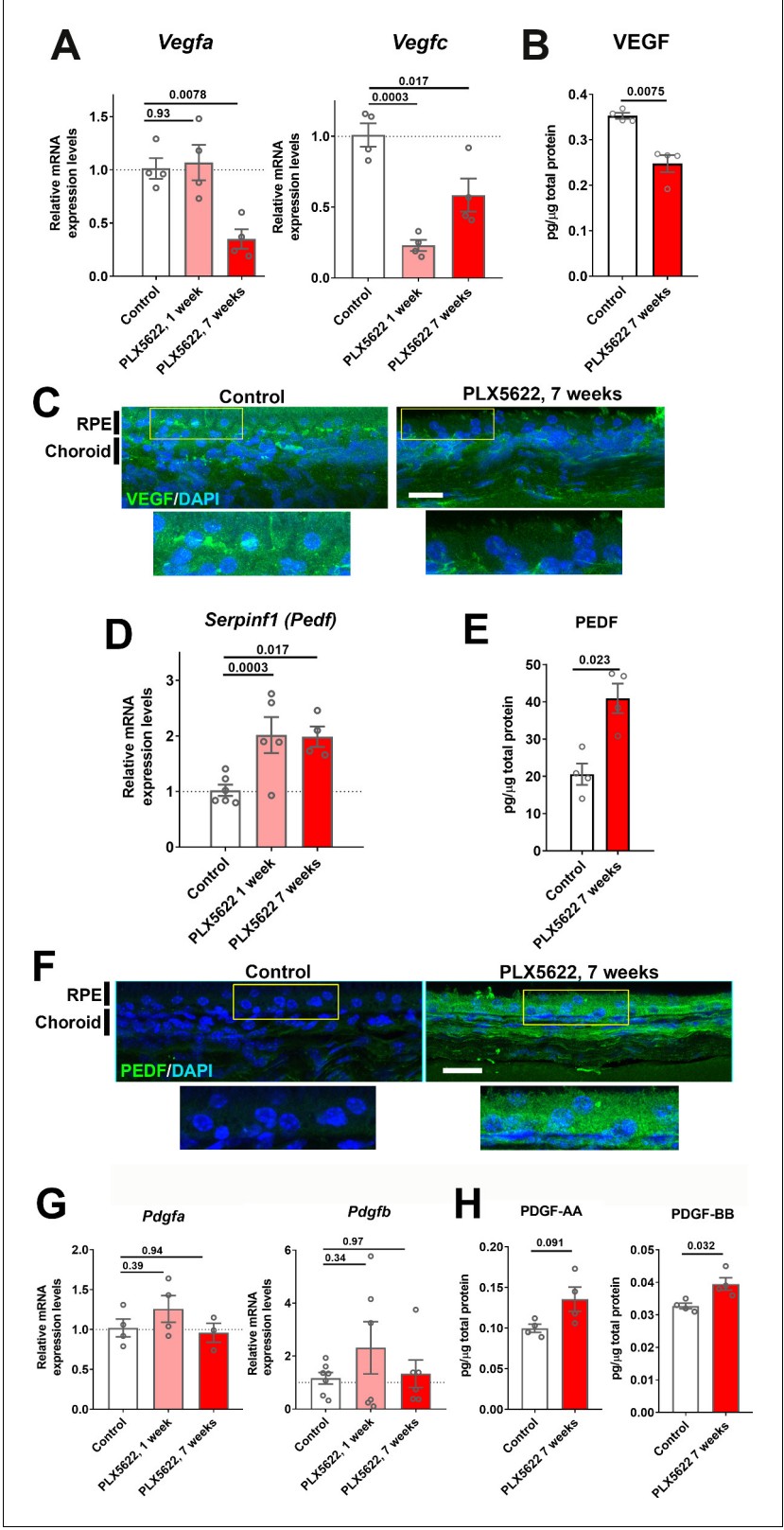

**Figure 5.** Expression levels of angiogenic growth factors in the RPE-choroid complex are altered with long-term choroidal macrophage depletion. Total mRNA and protein were isolated from RPE-choroid complexes of animals treated with PLX5622-containing diet for 1 or 7 weeks and untreated control animals. mRNA levels of *Vegfa* and *Vegfc* (**A**), and protein level of VEGF (**B**) were all significantly reduced following 7 weeks of depletion.
*Figure 5 continued on next page*

*Figure 5 continued*

Immunohistochemical localization demonstrated VEGF expression at the level of the RPE that was decreased following macrophage depletion (C). *Serpinf1* mRNA (D) and PEDF protein (E) were increased following macrophage depletion; immunohistochemical analyses demonstrated increased PEDF immunopositivity in the RPE layer and choroid (F). *Pdgfa* and *Pdgfb* mRNA levels (G) and PDGF-AA and PDGF-BB protein levels (H) were largely unaltered with macrophage depletion. P values were from a 1-way ANOVA, n = 4 to 8 animals per treatment group.

The online version of this article includes the following figure supplement(s) for figure 5:

**Figure supplement 1.** mRNA expression levels of Tgfb1 were altered following sustained depletion of choroidal macrophages.

**Figure supplement 2.** Secondary antibody staining control.

continuously depleted animals showed a corresponding shift towards an increased prevalence of larger cells at 7 weeks depletion, those in the depletion-repopulation group were similar to those in untreated controls (*Figure 8C*). Further, macrophage repopulation also prevented the increase of multinucleated cells ($\geq$3 nuclei) induced with 7 weeks depletion (*Figure 8D*). These findings indicated that structural changes in RPE cells, a phenotype of later onset apparent at 7 weeks but not at 3 weeks of depletion, can be prevented if the choroidal macrophage population were reinstated by repopulation.

The effects of macrophage repopulation on RPE structure were also observed in analyses of RPE gene expression. Relative to the continuous depletion group, the depletion repopulation group showed amelioration in depletion-induced decreases in mRNA (*Figure 8—figure supplement 1A*) and protein (*Figure 8E*) expression of RPE65, and mRNA expression of visual cycle genes (*Lrat* and *Rlbp1*), *Mitf*, and *Tjp1/Zo1* (*Figure 8—figure supplement 1A–C*). In addition to the RPE morphology and gene expression changes, the scotopic a- and b-wave amplitudes decreased significantly at 3 weeks of depletion in the continuous depletion group and decreased slightly further in the scotopic b-wave amplitude at 7 weeks of depletion while the scotopic a-wave amplitude remained stable. However, in the depletion-repopulation group, while similar decrease in the scotopic a- and b-wave amplitudes were observed at 3 weeks of depletion, the scotopic a- and b-wave amplitudes were unchanged between 3 to 7 weeks during macrophage repopulation (*Figure 8—figure supplement 2A*). Depletion-induced reductions in *Vegfa* and *Vegfc* mRNA and VEGF protein expression were largely reversed with repopulation (*Figure 9A*), while there was a trend towards a reversal of depletion-induced increases of *Serpinf1* (*Pedf*) mRNA and PEDF protein expression (*Figure 9B*). Taken together, these data demonstrate that structural and gene expression changes induced in the RPE-choroid complex upon macrophage depletion were significantly ameliorated or stabilized upon macrophage repopulation, supporting the notion of a homeostatic supportive function for choroidal macrophages.

## Discussion

Tissue-resident innate immune cells, in addition to playing multiple inflammation-related functional roles, have been recently recognized to carry out constitutive homeostatic functions in adult tissues (*Wynn et al., 2013*). In the adult brain, these cell populations comprise of microglia, which are located behind the blood-brain barrier (BBB) within the brain parenchyma (*Wolf et al., 2017*), and border-associated macrophages, which are located external and adjacent to the BBB in the meninges and choroid plexus (*Van Hove et al., 2019*). How these populations mediate CNS homeostasis under normal conditions is an active area of investigation (*Li and Barres, 2018*). In the eye, microglia resident within the healthy retina help maintain the structural and functional integrity of neuronal synapses (*Wang et al., 2016*), but the role of resident macrophages within the choroid of the eye is less well understood (*McMenamin et al., 2019*). Significantly, deficiencies in maintaining long-term structural homeostasis of the choroid feature importantly in ocular pathologies; the vascular choroid in in clinical studies has been found to undergo progressive atrophic thinning with normal aging (*Abbey et al., 2015*; *Wakatsuki et al., 2015*; *Zhang et al., 2014*) and with increasing severity of AMD (*Keenan et al., 2020*), which are supported by histopathological and molecular evidence of vascular dropout and attenuation in these scenarios (*Bhutto and Lutty, 2012*; *Chirco et al., 2017*;

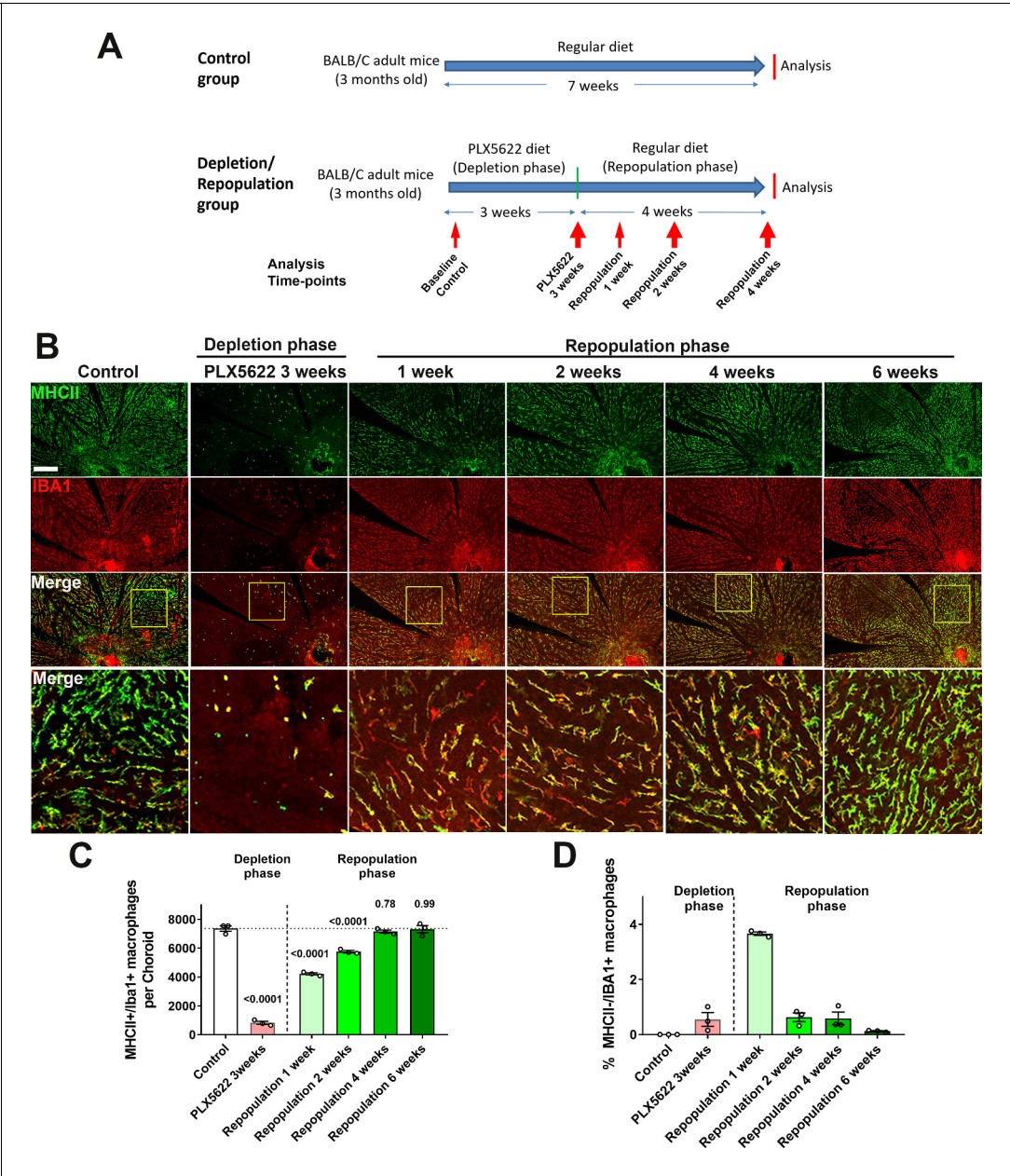

**Figure 6.** Choroidal macrophages demonstrate spontaneous repopulation and recovery of constitutive numbers following depletion. (**A**) Adult 3 month old mice were administered a diet containing PLX5622 for 3 weeks to achieve depletion of choroidal macrophages (depletion phase), and then returned to a standard diet for another 4 weeks (repopulation phase). The control group consisted of age-matched animals maintained on a standard diet. (**B**) Sclerochoroidal flatmounts were immunolabeled with MHCII (*green*) and IBA1 (*red*) to visualize the resident macrophages. Panels show representative images and yellow box marking the areas showing high-magnification views. Scale bar = 1 mm. (**C**) The total number of macrophages in the choroid that were immunopositive for both IBA1 and MHCII markedly decreased after 3 weeks of PLX5622 administration but progressively increased following PLX5622 cessation to reach baseline levels after 4–6 weeks. (**D**) While all IBA1+ macrophages were immunopositive for MHC at baseline, a small fraction of IBA1+ macrophages were immunonegative for MHCII early in the repopulation phase (at 1 week following resumption of a standard diet). This phenotype was transient as all IBA1+ macrophages reacquired MHCII immunopositivity when repopulation neared completion at 6 weeks of repopulation. P values were from a 1-way ANOVA, n = 3 animals per treatment group.

The online version of this article includes the following figure supplement(s) for figure 6:

**Figure supplement 1.** Repopulating choroidal macrophages demonstrate proliferation in situ and progressive morphological maturation.

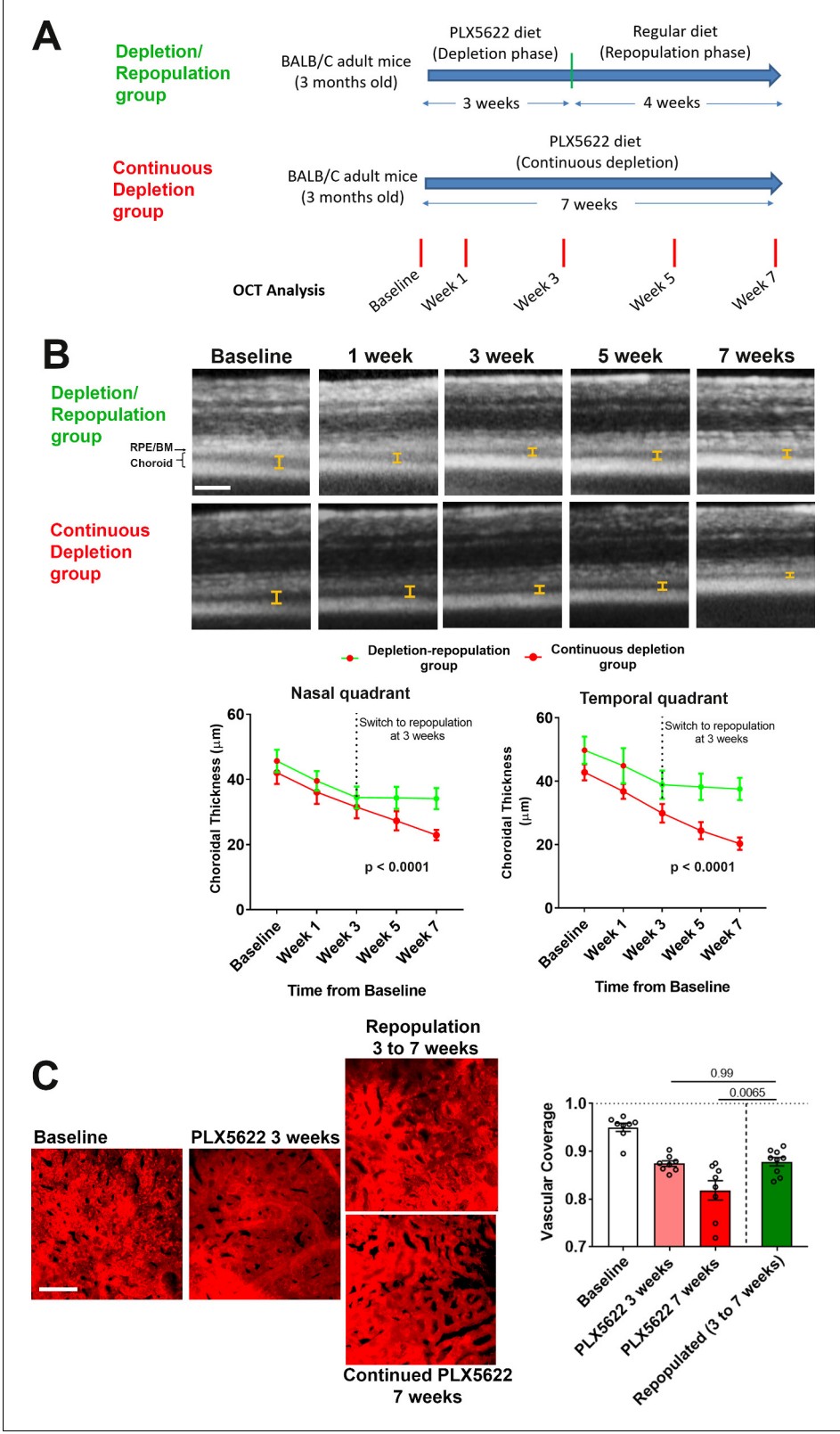

**Figure 7.** Choroidal macrophage repopulation following depletion is accompanied by an arrest in depletion-associated choroidal vascular atrophy. (**A**) Experimental plan for the assessment of vascular changes in the choroid during macrophage depletion and repopulation. Age-matched 3 month-old adult BALB/cJ animals were divided into two groups: (1) the depletion-repopulation group: animals administered PLX5622-containing diet for 3 weeks,

*Figure 7 continued on next page*

*Figure 7 continued*

followed by 4 weeks on a standard diet, and (2) the continuous depletion group: animals were administered PLX5622-containing diet continuously for 7 weeks. Animals in both groups were assessed at baseline, and 1, 3, 5, and 7 weeks following baseline with serial OCT imaging. (B) Representative longitudinal aligned in vivo EDI OCT images of animals in the two groups taken at different time points; bars show the intervals across which choroidal thickness were measured. Scale bar = 200 μm. Choroidal thicknesses measured in the temporal and nasal quadrants (at 300 μm temporal and nasal respectively from the optic nerve at the horizontal meridian) decreased progressively with time in the continuously depleted group (*red* line), but stabilized in the depletion/repopulation group (*green* line) upon the onset of repopulation at 3 weeks following baseline, n = 9 eyes from five animals in each group for all time-points, p values indicate comparisons between the two experimental groups, 2-way ANOVA using a mixed-effects analysis. (C) Choriocapillaris vascular structure was imaged in sclerochoroidal flatmounts labeled with systemic DiI perfusion. Scale bar = 50 μm. The fractional area coverage by choriocapillaris vessels decreased continuously up to 7 weeks following baseline in the continuous depletion group but was stabilized from 3 to 7 weeks following baseline in the depletion/repopulation group. P values were from a 1-way ANOVA, n = 8 eyes from four animals per treatment group.

*Seddon et al., 2016*; *Sohn et al., 2019*). Previous studies have associated AMD severity with increased choroidal macrophage number and activation (*McLeod et al., 2016*), however whether age-related and AMD-related choroidal changes may arise from deficient homeostatic function of resident macrophages has not been previously explored.

In the current work, we discover that choroidal macrophages may indeed play a previously unappreciated trophic role in maintenance of the vascular structure of the choroid. We found that the ablation of a significant majority of choroidal macrophages was accompanied by a generalized and progressive atrophy of the choroidal vasculature affecting choroidal vessels at multiple levels. This atrophic process extended to multiple levels of the vascular choroid, resulting in an overall decrease in choroidal thickness and choriocapillaris attenuation at the border of the RPE-Bruch's membrane interface. The ability of resident innate immune cells to maintain local vasculature appears to vary according to tissue context; within the CNS, the depletion of retinal or brain microglia results induces changes in vessel structure and permeability in developing systems (*Arnold and Betsholtz, 2013*; *Stefater et al., 2011*), but not in adult systems (*Elmore et al., 2014*; *Wang et al., 2016*). In the adult gut, the depletion of resident macrophage subsets results in vascular alterations in the submucosa (*De Schepper et al., 2018*), a feature observed analogously in the adult choroid. In the choroid, resident macrophages are distributed in a manner suitable for vascular regulation; they are found in close proximity to choroidal vessels at all levels and possess a dendritiform ramified morphology with dynamic cellular processes that constitutively survey the perivascular space (*Kumar et al., 2014*; *McMenamin, 1999a*). As such, choroidal macrophages may function to integrate cues from the local environment to shape vascular organization and blood supply, adapting to ongoing physiological needs in the outer retina. Interestingly, the phenotype of choroidal vascular attenuation that accompanied macrophage ablation resembled that occurring with aging in which vascular structural losses were accompanied by a general deramification of choroidal macrophages and thus a potential decrease in macrophage-vasculature interactions. This similarity supports the notion that macrophages can signal endogenously to the vascular choroid in ways that facilitate their constitutive maintenance.

We observed here that the trophic influence of choroidal macrophages may extend also to the maintenance of RPE structure and function. The nature of RPE changes accompanying macrophage depletion has a notable resemblance to structural and functional alterations seen with aging and AMD in the RPE layer, including increased cellular disorganization and decreased cell density in the RPE monolayer (*Bhatia et al., 2016*; *Del Priore et al., 2002*), increased numbers of multinucleated cells (*Chen et al., 2016*), and increased RPE dedifferentiation (*Makarev et al., 2014*). Previous studies have connected microglia/macrophage activation with RPE alterations, but these had been in pathological situations in which increased production of inflammatory mediators induces RPE cell changes (*Kutty et al., 2016*; *Ma et al., 2009*; *Yamawaki et al., 2016*). Contrastingly, our findings here suggest that trophic macrophage-RPE signaling may be constitutively present under normal conditions, indicating that not only macrophage overaction, but also insufficiency of endogenous macrophage function, may contribute pathogenically to changes in the choroid and RPE layer observed in age-related retinal degeneration.

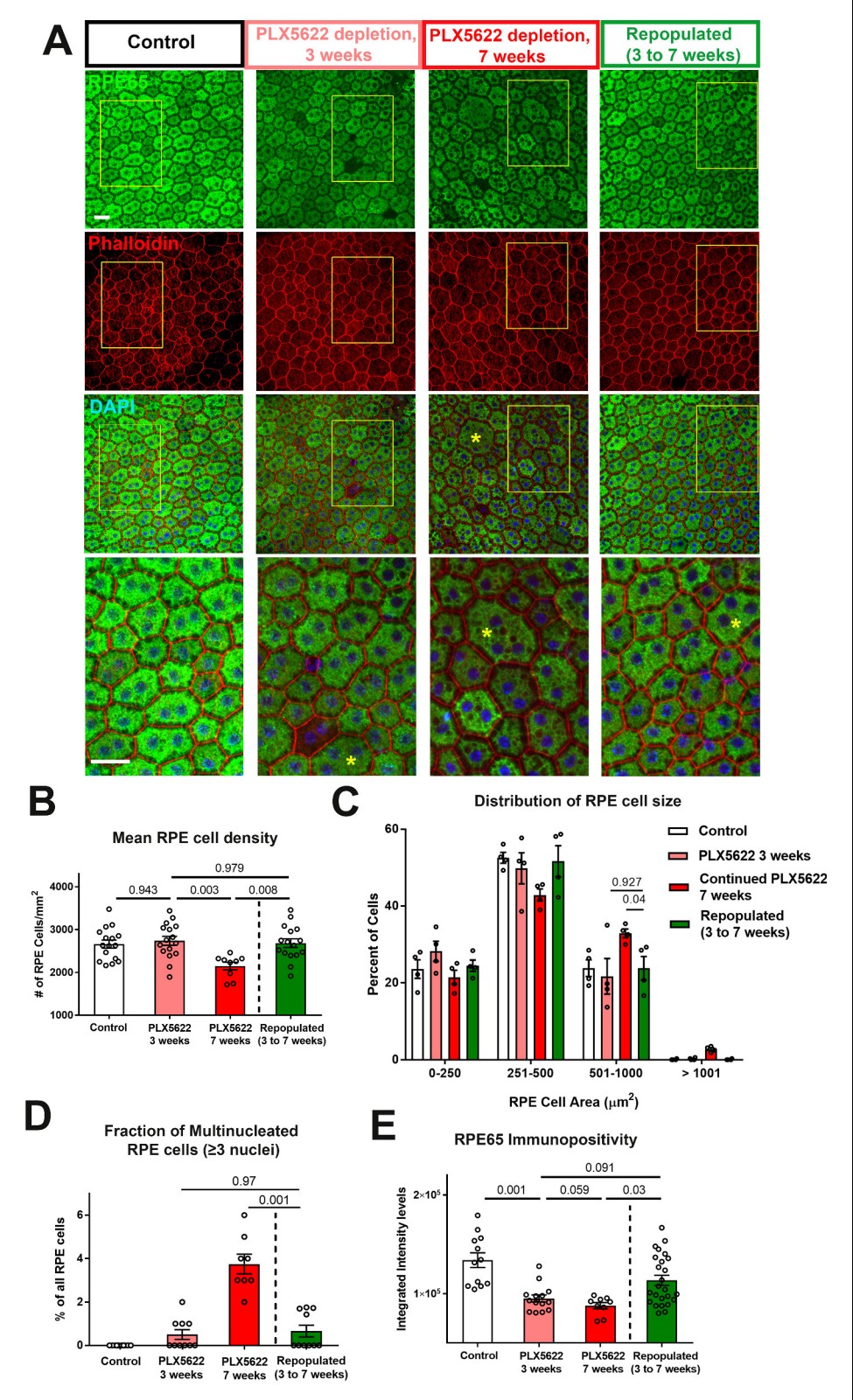

**Figure 8.** Choroidal macrophage repopulation following depletion is accompanied by an arrest in depletion-associated alterations in RPE structure and gene expression. (**A**) Immunohistochemical analyses of the RPE-sclerochoroidal flatmounts from (1) untreated control animals (*white* bars), (2) animals continuously treated for 3 weeks of PLX5622-containing diet (*pink* bars), (3) animals continuously treated for 7 weeks of PLX5622-containing diet (*red* bars), and (4) animals treated for 3 weeks of PLX5622 containing diet and then followed by 4 weeks of standard diet (*green* bars); were performed with

*Figure 8 continued on next page*

*Figure 8 continued*

RPE65 (*green*), conjugated phalloidin (*red*), and DAPI (*blue*). The progressive patchy decrease in RPE65 immunopositivity and increase of large multinucleated (≥3 nuclei) cells were observed in animals in the 3- and 7 weeks depletion groups, however these changes were recovered upon the onset of repopulation. High-magnification views in the inset (yellow boxes) demonstrated the presence of large cells (>500 $\mu m^2$ in area, asterisks) with multiple nuclei in the 7 week depleted group but was less prominent in the repopulated group. (B) Quantitative analysis of RPE morphology revealed that repopulation of choroidal macrophages after 3 weeks of depletion prevented further decrease in RPE cell density evident on 7 weeks of depletion. (C) Analysis of the distribution of RPE cell areas revealed that the increase in the proportion of large cells (>500 $\mu m^2$ in area) induced with 7 weeks depletion was prevented in the repopulation group, which resembled the control and 3 week depletion groups in their distribution. (D) Analysis of the proportion of multinucleated RPE cells (≥3 nuclei) showed that the increased proportion of multinucleated cells induced with 7 weeks depletion was prevented in the repopulation group. (E) Quantitation of mean intensity of RPE65 immunopositivity showed that the significant decrease in RPE65 staining induced by depletion was partially restored after macrophage repopulation. P values indicate comparisons computed from a 1-way ANOVA, n = analysis of 4 imaging field from each animal, four animals per treatment group.

The online version of this article includes the following figure supplement(s) for figure 8:

**Figure supplement 1.** Macrophage depletion-associated changes in RPE genes expression are ameliorated with macrophage repopulation.
**Figure supplement 2.** Changes in ERG amplitudes during macrophage depletion and subsequent macrophage repopulation.

In our experiments here, we have employed pharmacological CSF1R inhibition to ablate choroidal macrophages to evaluate their agency in choroidal homeostasis. PLX5622 had been designed to target CSF1R with high specificity, and demonstrates 200-fold higher selectivity over KIT and FLT3, the two most homologous receptors of CSF1R (*Spangenberg et al., 2019*), indicating improvements over earlier less selective precursors (e.g. PLX3397) and decreasing the likelihood of off-target effects. While other methods exist to ablate innate cell populations (*Spangenberg and Green, 2017*), longer-term sustained ablations such as those featured in our experiments are currently possible only with pharmacological CSF1R inhibition. While it cannot be ruled out that PLX5622 may have induced off-target effects that produce the phenotypes observed here independently of CSF1R inhibition and macrophage depletion, there has not been evidence from the multiple studies involving its use in which deleterious effects, including vascular degenerative phenotypes, have been detected in the brain (*Dagher et al., 2015*), spinal cord (*Halder and Milner, 2019*), or retina (*Zhang et al., 2018*) in non-disease states. In our previous work, we had noted that CSF1R inhibition using PLX5622 additionally ablates microglia in the healthy adult retina and that neuronal synapses in the plexiform layers in close proximity to retinal microglia showed structural and functional degeneration (*Zhang et al., 2018*). While we have related choroidal vascular and RPE changes to the ablation of choroidal macrophages, which are in close physical proximity and have been shown to be capable of exerting their influence, we cannot rule out the possibility that the ablation of more distant microglia in the inner retina may also contribute to these associated phenotypes, although this 'action-from-a-distance' capability have not been previously demonstrated.

Our findings here indicate the existence of cellular and molecular mechanisms that may underlie trophic relationships between choroidal macrophages, choroidal vasculature, and RPE cells. Multiple studies have shown that the maintenance of vasculature in the adult choroid is highly dependent on the nearby presence of viable RPE cells, and specifically on RPE-expressed angiogenic factors (*Blaauwgeers et al., 1999*; *Saint-Geniez et al., 2006*). Measures that ablate RPE cells (*Hayashi et al., 1999*; *Korte et al., 1984*), or otherwise deplete or prevent the delivery of RPE-derived VEGF (*Marneros et al., 2005*; *Saint-Geniez et al., 2009*), resulted in prominent atrophy of choroidal vessels. This dependence of the choroidal vasculature on constitutive RPE support indicates that it is possible that choroidal macrophages may contribute to vascular maintenance indirectly via their effects on RPE cells (*Figure 9—figure supplement 1*). In our experiments, we observed that macrophage depletion is accompanied by a significant decrease in ratio of VEGF/PEDF factors that are expressed in the RPE-choroidal complex, with a significant decrease in VEGF immunopositivity and an increase of PEDF in the RPE cells layer, indicating that macrophage depletion may induce vascular atrophy by decreasing RPE-mediated angiogenic support. One mode of macrophage-to-RPE signaling that is potentially relevant in this mechanism involves TGFβ; TGFβ ligands, particularly TGFB1, is prominently expressed by tissue macrophages (*Huen et al., 2013*; *Kwong et al., 2006*). Signaling via TGFBR1/2, which is expressed in RPE cells, has been shown to potentiate RPE production and secretion of VEGF (*Bian et al., 2007*; *Nagineni et al., 2003*). We observed that *Tgfb1* expression is significantly decreased in the RPE-choroidal complex with

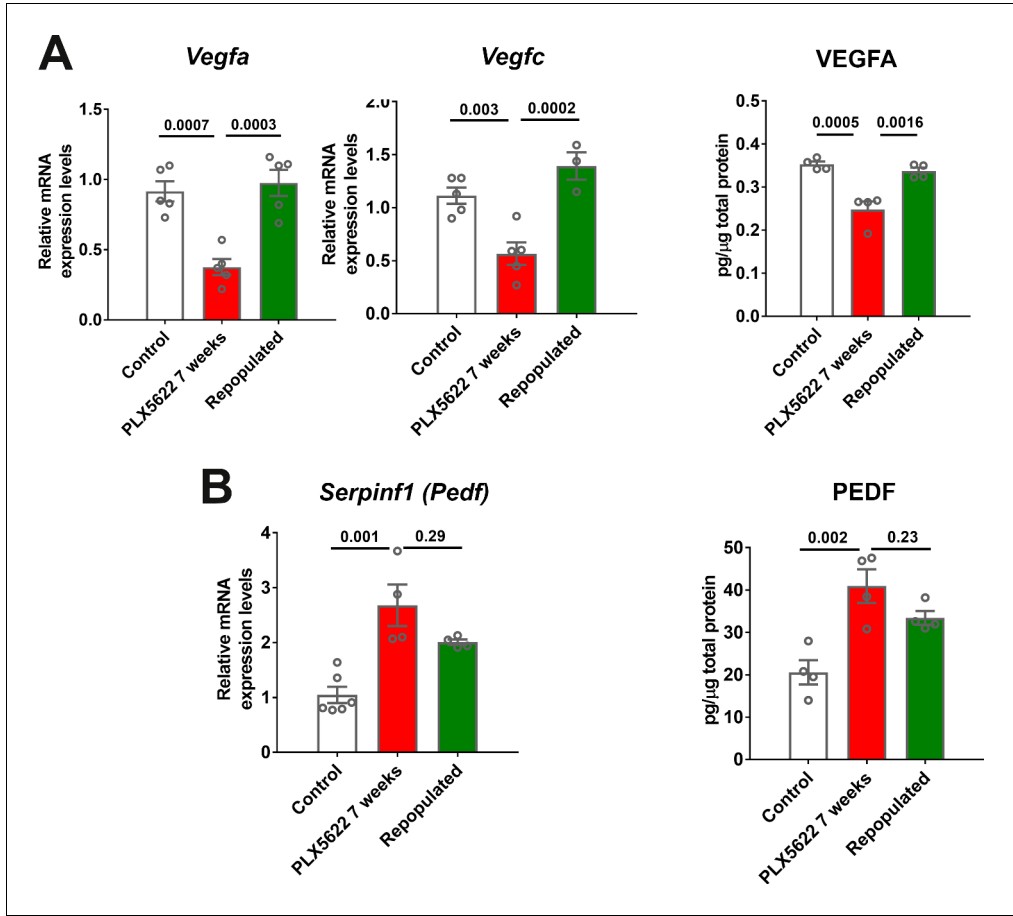

**Figure 9.** Choroidal macrophage repopulation following depletion is accompanied by a reversal in angiogenic factor expression. mRNA and total protein were isolated from the RPE-choroid complex of (1) control animals maintained on a standard diet (*white* bars), (2) depleted animals treated continuously with a PLX5622-containing diet for 7 weeks (*red* bars), and (3) repopulated animals treated with PLX5622-containing diet for 3 weeks and then switched to a standard diet for 4 weeks (*green* bars). mRNA and protein levels of angiogenic factors were analyzed by qrt-PCR and multiplex assay/ELISA respectively. (**A**) mRNA levels of *Vegfa* and *Vegfc*, and protein level of VEGFA, which were all decreased in the depleted group relative to the control group, were recovered back to control levels in the repopulated group. (**B**) mRNA levels of *Serpinf1*, and protein levels of PEDF, which were increased in the depleted group relative to the control group, were found at intermediate levels in the repopulated group. P values were from a 1-way ANOVA, n = 4 to 6 animals per treatment group.
The online version of this article includes the following figure supplement(s) for figure 9:

**Figure supplement 1.** Schematic showing putative signaling between choroidal macrophages, vasculature, and the RPE layer in the outer retina under normal conditions and with choroidal macrophage depletion.

choroidal depletion, suggesting that TGFB1 secretion by choroidal macrophages to TGBR1/2 on RPE cells may contribute here. Verification of this mode of signaling, as well as other potential modes, await studies involving inducible macrophage-specific loss-of-function interventions in suitable in vivo models. Alternatively, it is possible that macrophage-vasculature signaling may occur directly, and that the RPE structural changes observed here may be secondarily induced by a resulting deficiency of choroidal supply. VEGF-secreting choroidal macrophages can directly potentiate choroidal neovascularization, as demonstrated in experimental studies (*Seddon et al., 2016*) and suggested by macrophage presence in neovascular membranes in exudative AMD (*Cherepanoff et al., 2010*). However, our immunohistochemical data do not indicate that macrophages in the healthy choroid constitutively secrete significant levels of VEGF, unlike those in neovascular membranes (*Grossniklaus et al., 2002*; *Ishibashi et al., 1997*). Future studies identifying

constitutive angiogenic factors expressed by macrophages in the healthy choroid will help address this mechanism of direct macrophage-vascular signaling.

We additionally discover here that macrophages in the adult choroid, which under normal conditions are non-migratory and demonstrate little constitutive proliferation (*Kumar et al., 2014*), can be induced by depletion to initiate a regenerative program de novo, in ways akin to barrier-associated macrophages in the brain (*Van Hove et al., 2019*). The cellular source for repopulating macrophages in this context is yet undetermined, and may involve the proliferation of residual resident macrophages, as observed for microglia in the adult retina (*Zhang et al., 2018*), or the infiltration of circulating monocytes, as occurs in models of retinal injury (*Ma et al., 2017*). Our observations here highlight that the regeneration process involves the in situ proliferation of repopulating cells in the choroid, morphological and marker maturation during repopulation, and an accurate recapitulation of original macrophage density, distribution, and morphology, akin to the repopulation of microglia in the retina (*Zhang et al., 2018*) and brain (*Elmore et al., 2014*). The ability of regenerated macrophages to recapitulate the endogenous function of the original macrophages is supported here by the arrest of choroidal vascular atrophy and RPE degenerative change, and the recovery of angiogenic factor balance upon the full restoration of macrophage numbers in the choroid. Interestingly, we did not observe a complete reversal of vascular and RPE degenerative changes that recovers the original, pre-depletion anatomy in the RPE-choroid, suggesting that aspects of degenerative change may be permanent and resist restoration by reinstituted trophic signals. This ability to regenerate choroidal macrophages presents a new potential therapeutic opportunity for immune modulation in the choroid in which depletion-repopulation measures may be employed to 'reset' the immune environment in the choroid in pathological situations in which choroidal macrophages demonstrate chronic activation. The feasibility of this approach will require further elucidation of the nature of macrophage repopulation in models of retinal pathology.

Taken together, our results here have significant relevance to understanding choroidal degeneration in aging and AMD progression (*Arya et al., 2018*) and to ongoing translational efforts to preserve or restore the vascular choroid in age-related disease (*Whitmore et al., 2015*). Additional to the concept that abnormal and imbalanced immune activation within the choroid involving proinflammatory and complement-mediated mechanisms can exert deleterious effects on choroidal vascular structure, our findings show that non-inflammatory trophic influences from resident choroidal macrophages may be additionally important in maintaining choroidal structural integrity across the lifespan of the organism. Our findings suggest that age- and AMD-associated degeneration in the RPE-choroid complex may be contributed to by multiple immune influences that combine inappropriate inflammatory activation and insufficient trophic support. The balance between these possibly simultaneous mechanisms may influence the relative severity of degeneration between the choriocapillaris vs. the adjacent RPE layer, which may influence the progression to the atrophic vs. the neovascular forms of AMD as hypothesized from histopathological studies (*Seddon et al., 2016*). As such, immunomodulatory interventions aimed at preserving the integrity of the RPE-choroid complex should optimally enable both increased macrophage trophic influences and decreased inflammatory damage. These perspectives add to the strategy of choroidal macrophage-targeted interventions as potential therapies aimed at choroidal preservation in diseases like AMD.

## Materials and methods

### Key resources table

| Reagent type (species) or resource | Designation | Source or reference | Identifiers | Additional information |
|---|---|---|---|---|
| Antibody | anti-Iba1 Rabit polyclonal | Wako | Cat# 019–19741 RRID:AB_839504 | Dilution 1:200 |
| Antibody | anti-MHC-II Rat monoclonal | BD Pharmingen | Cat# 556999 RRID:AB_396545 | Dilution 1:200 |
| Antibody | anti-RPE65 Mouse monoclonal | Millipore | Cat# MAB5428 RRID:AB_571111 | Dilution 1:200 |

*Continued on next page*

Continued

| Reagent type (species) or resource | Designation | Source or reference | Identifiers | Additional information |
|---|---|---|---|---|
| Antibody | anti-VEGFa Rabit polyclonal | Abcam | Cat# ab46154 RRID:AB_2212642 | Dilution 1:200 |
| Antibody | anti-PEDF Rabit polyclonal | Abcam | Cat# ab180711 RRID:AB_2827998 | Dilution 1:200 |
| Commercial assay or kit | Mouse SERPINF1/ PEDF ELISA kit | LifeSpan BioSciences | LS-F36110 | |
| Commercial assay or kit | Magnetic Luminex Assay kit for VEGF, PDGF-AA and PDGF-BB | R and D Systems | LXSAMSM-11 | |
| Sequence-based reagent | Vegfa_F | IDT | PCR primers | TGTGCGCAGACAGTGCTCCA |
| Sequence-based reagent | Vegfa_R | IDT | PCR primers | CCTGGGACCACTTGGCATGG |
| Sequence-based reagent | Vegfc_F | IDT | PCR primers | GGGAAATGTGCCTGTGAATG |
| Sequence-based reagent | Vegfc_R | IDT | PCR primers | GTTCAGATGTGGCCTTTTCC |
| Sequence-based reagent | Serpinf1_F | IDT | PCR primers | CACCCAAGTGGAACACAGG |
| Sequence-based reagent | Serpinf_R | IDT | PCR primers | TTAAGTACTACTGGGGTCCA |
| Sequence-based reagent | Rpe65_F | IDT | PCR primers | GCCAATTTACGTGAGAATTGGG |
| Sequence-based reagent | Rpe65_R | IDT | PCR primers | CAGTCCATGGAAGGTCACAG |
| Sequence-based reagent | Lrat_F | IDT | PCR primers | CCGTCCCTATGAAATCAGCTC |
| Sequence-based reagent | Lrat_R | IDT | PCR primers | ATGGGCGACACGGTTTTCC |
| Sequence-based reagent | Rlbp1_F | IDT | PCR primers | GGCACTTTCCGCATGGTT C |
| Sequence-based reagent | Rlbp1_R | IDT | PCR primers | CCGGGTCTCCTCCTTTTCAT |
| Sequence-based reagent | Mitf_F | IDT | PCR primers | CAGCCATAAACGTCAGTGTGC |
| Sequence-based reagent | Mitf_R | IDT | PCR primers | GAGTGAGCATAGCCATAGGGC |
| Sequence-based reagent | Tjp1_F | IDT | PCR primers | GAGAAAGGTGAAACTCTGCTG |
| Sequence-based reagent | Tjp1_R | IDT | PCR primers | GTGGTCAATCAGGACAGAAAC |
| Sequence-based reagent | Pdgfa_F | IDT | PCR primers | GCCAGCCTTCACGGGTCC |
| Sequence-based reagent | Pdgfa_R | IDT | PCR primers | CCTCACATCTGTCTCCTCCT |
| Sequence-based reagent | Pdgfb_F | IDT | PCR primers | CTGCAAGTGTGAGACAGTAG |
| Sequence-based reagent | Pdgfb_R | IDT | PCR primers | CTAGGCTCCGAGGGTCTC |
| Sequence-based reagent | Gapdh_F | IDT | PCR primers | GCCGCCTGGAGAAACCTGCCAA |
| Sequence-based reagent | Gapdh_R | IDT | PCR primers | GGGGTGGGTGGTCCAGGGTTT |

## Experimental animals

Experiments were conducted according to protocols approved by the National Eye Institute Animal Care and Use Committee and adhered to the Association for Research in Vision and Ophthalmology statement for the use of animals in ophthalmic and vision research. Animals were housed in a National Institutes of Health (NIH) animal facility under a 12 hr light/12 hr dark cycle with food ad libitum.

Adult 3 month-old female wild type BALB/cJ mice (The Jackson Laboratory, Bar Harbor, ME) and 3 month-old albino transgenic mice, containing a transgene with a α-smooth muscle actin (αSMA) promoter driving the expression of green fluorescent protein in perivascular cells in the choroid (*Condren et al., 2013*), were used to analyze cellular changes in the choroid and the RPE following macrophage depletion.

## Model for macrophage depletion-repopulation in the choroid

We employed a model for macrophage depletion in the adult mouse choroid which involved the dietary administration of PLX5622 (Plexxikon, Berkeley, CA), a potent and selective inhibitor of the CSF1R that we previously demonstrated to deplete microglia in the retina (*Zhang et al., 2018*). Animals were placed on a rodent chow containing PLX5622 (at 1200 parts per million, formulated by Research Diets Inc New Brunswick, NJ) for up to 7 weeks to induce depletion macrophages resident in the choroid. To allow for the repopulation of choroidal macrophages, animals subjected to depletion with PLX5622-containing diet for 3 weeks were switched back to a standard diet; the first day of control diet resumption was designated day 0 of the repopulation phase.

## Immunohistochemical analysis of sclerochoroidal tissue

Mice were euthanized by $CO_2$ inhalation. Enucleated eyes were dissected to form posterior segment eyecups which were then fixed in 4% paraformaldehyde in PBS for 1 hr at room temperature. Eyecups were either processed for vibratome sectioning (100 μm thick sections, VT1000, Leica) or dissected to obtain sclerochoroidal flatmounts, which were then immersed for 1 hr in a blocking buffer (1 × PBS containing 10% normal goat serum and 0.5% Triton X-100) at room temperature. Primary antibodies, which included IBA1 (1:200; Wako), MHC-II (1:200; BD Pharmingen), RPE65 (1:200; Millipore), VEGFA (1:200; Abcam), or PEDF (1:200; Abcam) were diluted in blocking buffer and added to sections/flat-mounts to incubate for 24 hr at 4°C. After washing in 1 × PBS with 0.2% Triton X-100, sections/flat-mounts were then incubated overnight with secondary antibodies (Alexa Fluor 488– or 568–conjugated anti-rabbit, rat or mouse IgG). Phalloidin (1:500; Invitrogen) was added and incubated for 4 hr label F-actin that enables visualization of RPE cell morphology. DAPI (1:500; Sigma-Aldrich) was added to label cell nuclei. Stained samples were imaged with confocal microscopy (Zeiss LSM700). To visualize macrophage distributions over the entire choroid, tiled images of sclerochoroidal flatmounts were obtained under a 10x objective and stitched together with image analysis software (Zen, Carl Zeiss Microscopy). For visualization of RPE monolayer, multiplane z-series were captured under a 40x oil-immersion objective. Resulting images were subjected to computer-assisted analysis (ImageJ, NIH); the density of choroidal macrophages was assessed across the entire choroid and expressed as number per unit area. The number of nuclei per RPE cells were manually assessed from high-magnification images of RPE flat-mounts captured at or near the midpoint between the optic nerve and the peripheral edge of the flat-mount.

## Automated RPE morphological analysis

Sclerochoroidal flatmounts immunolabeled with RPE65 and phalloidin were imaged with confocal microscopy (Zeiss LSM880) and multiplane z-series images were captured under a 40x oil-immersion objective. Morphological analyses of RPE mosaics were performed in an automated manner using REShAPE software, as previously described (*Sharma et al., 2019*). Briefly, individual RPE cells were segmented at their borders using MATLAB and a binary 'mask' has been generated to measure cell morphological features. RPE cell density was calculated as cell numbers per unit area and the area of individual RPE cells was determined from cell segmentation.

### In vivo optical coherence tomography (OCT) imaging of the choroid

Mice were anesthetized with intraperitoneal ketamine (90 mg/kg) and xylazine (8 mg/kg) and their pupils were dilated. Choroidal and retinal structure was assessed using OCT imaging (Spectralis, Heidelberg Engineering) in anesthetized animals through a dilated pupil. OCT images in the enhanced depth imaging (EDI)-mode were captured longitudinally in individual experimental animals in the same matched fundus location (with images captured using 'follow-up' mode in the instrument software). The imaging was performed using a 30x objective (Heidelberg) and scan fields, measuring 6.5 by 4.4 mm and centered on the optic nerve (1000 A-scans/horizontal B-scan, 25 horizontal B-scans, average of three frames per B-scan, each spaced 182 µm apart) were captured in each imaging session. Repeat OCT imaging was performed at 1, 3, 5, and 7 weeks following the initiation of PLX5622. Thickness measurements in OCT images were made using the instrument manufacturer software; choroidal thickness measurements (defined as the axial distance from the Bruch's membrane to the outer edge of the choroid) were measured at 300 µm temporal and nasal from the optic nerve on the horizontal meridian, and average retinal thickness measurements per eye (defined as the axial distance from the vitreal surface to the Bruch's membrane) were made at 300 µm temporal and nasal from the optic nerve in corresponding locations.

### Direct labeling and visualization of choroidal vessels

Choroidal vessels are labeled by cardiac perfusion of an aqueous solution containing 1,19-dioctadecyl-3,3,39,39-tetramethylindocarbocyanine perchlorate (DiI, D-282, Invitrogen/Molecular Probes, Carlsbad, CA), a lipophilic dye that incorporates into membranes of endothelial cells revealing the structure of choroidal vessels as previously described (Kumar et al., 2014). Briefly, experimental mice were euthanized by carbon dioxide inhalation and the thoracic cavity opened to expose the heart. A volume of 100 µL of DiI stock solution (6 mg/ml in 100% ethanol) was dissolved in 5 ml of diluent comprising 1xPBS and 5% glucose in a 1:4 ratio and the resulting solution delivered via cardiac perfusion at a rate of 1–2 mL/min. This was immediately followed by the perfusion of 6 mL of PBS. The animals were immediately enucleated and the retinal pigment epithelium (RPE)-sclera-choroid complex dissected from the harvested globes after fixation with 4% paraformaldehyde. These flat-mount preparations were mounted on glass slides in Fluoromount (Sigma-Aldrich) with the RPE cell layer uppermost. Confocal multiplane z-stack images of DiI-labeled endothelial cells were obtained with confocal microscopy (Zeiss LSM700) under a 63x oil-immersion objective at 300–600 µm temporal and nasal from the optic nerve. Images of labeled vessels in the choriocapillaris were used to compute the fraction of vascular coverage as the fractional area of the imaging field occupied by DiI-labeled vessels using ImageJ Software (NIH).

### Quantitative reverse transcription PCR analysis

Quantitative reverse transcription-PCR analysis of isolated mouse choroid was performed to identify the molecular changes after macrophage depletion. RPE-sclerochoroidal tissue were acutely isolated from euthanized animals and total RNA was extracted using a RNeasy Mini kit (Qiagen) and amplified to cDNA using the PrimeScript 1st strand cDNA synthesis kit (TaKaRa Bio) per manufacturers' instructions. cDNA in 1 µL volume was used for real-time PCR analysis using SYBR Green PCR Mastermix (Affymetrix) on the CFX96 Real-Time System (Bio-Rad). Levels of mRNA expression were normalized to those in controls as determined using the comparative CT ($2\Delta\Delta CT$) method. GAPDH was used as an internal control. Tgfb1 primer was purchased from Bio-Rad (Assay ID qMmuCID0017320), and other oligonucleotide primer pairs used are listed in the Key Resources Table.

### Histological analysis of Choroid-RPE sections

Eyes from control and 7 weeks PLX5622-treated animals were isolated and fixed for 30 min in 4% glutaraldehyde followed by 10% formalin for at least 24 hr. Fixed eyes were embedded in methacrylate, serially sectioned and stained with hematoxylin and eosin. Images were taken with a Zeiss microscope and mean thickness of the choroid from temporal and nasal quadrants and mean cross-sectional area of vascular lumina were measured using ImageJ software (NIH).

## Transmission Electron Microscopy of RPE sections

The eyes from control and PLX5622-treated mice were isolated and fixed in 2.5% glutaraldehyde. The specimens were prepared for transmission electron microscopy (TEM) as previously described (*Ogilvy et al., 2014*). Briefly, specimens were embedded in Spurr's epoxy resin and processed into ultrathin sections (90 nm) which were then stained with uranyl acetate and lead citrate and imaged with a transmission electron microscope (JEOL, JM-1010).

## Protein quantitation in choroid-RPE-complex

Mouse RPE-choroidal samples were isolated from the following groups: untreated control group, group treated with 7 weeks of PLX5622, group treated with 3 weeks of PLX5622 and switched to regular diet for another 4 weeks to allow macrophage repopulation. Four biological replicates were obtained for each group. Pigment epithelium-derived factor (PEDF) protein levels were measured in the samples using the Mouse SERPINF1/PEDF ELISA kit (LifeSpan BioSciences; LS-F36110). Protein levels of VEGF, PDGF-AA and PDGF-BB were analyzed using a Magnetic Luminex Assay kit (R and D Systems, LXSAMSM-11).

## Electroretinographic (ERG) analysis

ERGs were recorded using an Espion E2 system (Diagnosys). Mice were dark-adapted for 12 hr and their pupils were dilated prior to ERG assessment. Mice were anesthetized and flash ERG recordings were obtained simultaneously from both eyes using gold wire loop electrodes, with the reference electrode placed in the animal's mouth and the ground subdermal electrode at the tail. ERG responses were obtained at increasing light intensities over the ranges of $1 \times 10^{-4}$ to 10 cd·s/m$^2$ under dark-adapted conditions, and 0.3 to 100 cd·s/m$^2$ under a background light that saturates rod function. The stimulus interval between flashes varied from 5 s at the lowest stimulus strengths to 60 s at the highest ones. Two to ten responses were averaged depending on flash intensity. ERG signals were sampled at 1 kHz and recorded with 0.3 Hz low-frequency and 300 Hz high-frequency cutoffs. Analysis of a-wave and b-wave amplitudes was performed using customized Espion ERG Data Analyzer software (v2.2) that digitally filters out high-frequency oscillatory potential wavelets. The a-wave amplitude was measured from the baseline to the negative peak and the b-wave was measured from the a-wave trough to the maximum positive peak. ERGs were recorded before and after macrophage depletion. The a- and b-wave measurements of animals with or without macrophage depletion were compared using a two-way ANOVA.

## Statistical analysis

Statistical analyses were performed using Prism 7.0d (GraphPad). For comparisons involving two data columns, t tests (paired or unpaired) or nonparametric tests (Mann–Whitney U) were used, depending on whether the data followed a Gaussian distribution as determined by normality tests. A normality test (D'Agostino and Pearson) was used to analyze the distribution of all datasets. For comparisons involving three or more data columns, a one-way ANOVA (with Dunnett's multiple-comparisons test) was used if the data in an experimental group were found in a Gaussian distribution, and a nonparametric Kruskal–Wallis test (with Dunn's multiple-comparisons test) was used otherwise.

## Acknowledgements

This work is supported by National Eye Institute Intramural Research Program.

## Additional information

### Funding

| Funder | Grant reference number | Author |
| --- | --- | --- |
| National Eye Institute | Intramural Research Program | Xiao Yang<br>Lian Zhao<br>Maria M Campos<br>Mones Abu-Asab<br>Davide Ortolan<br>Nathan Hotaling<br>Kapil Bharti |

The funders had no role in study design, data collection and interpretation, or the decision to submit the work for publication.

### Author contributions

Xiao Yang, Conceptualization, Data curation, Software, Formal analysis, Validation, Investigation, Visualization, Methodology, Writing - original draft, Writing - review and editing; Lian Zhao, Davide Ortolan, Software, Formal analysis, Methodology, Writing - review and editing; Maria M Campos, Mones Abu-Asab, Resources, Methodology, Writing - review and editing; Nathan Hotaling, Software, Methodology, Writing - review and editing; Kapil Bharti, Resources, Supervision, Methodology, Writing - review and editing; Wai T Wong, Resources, Supervision, Funding acquisition, Writing - review and editing

### Author ORCIDs

Xiao Yang ⓘ http://orcid.org/0000-0003-0429-9188
Wai T Wong ⓘ https://orcid.org/0000-0003-0681-4016

### Ethics

Animal experimentation: This study was performed in strict accordance with the recommendations in the Guide for the Care and Use of Laboratory Animals of the National Institutes of Health. All of the animals were handled according to approved institutional animal care and use committee (IACUC) protocols (NEI-665) of the National Eye Institute.

### Decision letter and Author response

Decision letter https://doi.org/10.7554/eLife.55564.sa1
Author response https://doi.org/10.7554/eLife.55564.sa2

## Additional files

### Supplementary files

• Transparent reporting form

### Data availability

All data generated or analyzed during this study are included in the manuscript and supporting files.

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
