## [Decision Letter]

**Acceptance summary:**

Mechanisms that maintain choroidal homeostasis are not well understood, yet are important in aging and age-related macular degeneration (AMD). The authors report that ablation of choroidal macrophages via CSF1R blockade is associated with choroidal atrophy and retinal pigment epithelial (RPE) changes, decrease in visual cycle gene expression, and altered angiogenic factor expression. Suspending CSF1R blockade allowed macrophage regeneration, which arrested vascular degeneration and reversed RPE alterations. The study suggests that choroidal macrophages help maintain choroidal vasculature and RPE cells, and macrophages may be a factor in AMD pathology.

**Decision letter after peer review:**

Thank you for submitting your article "CSF1R blockade induces macrophage ablation and results in vascular atrophy and RPE disorganization in the mouse choroid" for consideration by *eLife*. Your article has been reviewed by three peer reviewers, including Lois Smith as the Reviewing Editor and Reviewer #1, and the evaluation has been overseen by Tadatsugu Taniguchi as the Senior Editor. The following individual involved in review of your submission has agreed to reveal their identity: Ye Sun (Reviewer #2).

The reviewers have discussed the reviews with one another and the Reviewing Editor has drafted this decision to help you prepare a revised submission.

Summary:

The choroid degenerates in aging and age-related macular degeneration (AMD). Mechanisms that maintain choroidal homeostasis are not well understood. The authors report that the ablation of choroidal macrophages via CSF1R blockade was associated with choroidal vascular atrophy and retinal pigment epithelial (RPE) structural disruption, down regulation of visual cycle genes, and altered angiogenic factor expression. Suspending CSF1R blockade then enabled spontaneous macrophage regeneration, which fully restored original macrophage distributions and morphologies. Macrophage regeneration arrested vascular degeneration and reversed RPE alterations. These findings suggest that choroidal macrophages help maintain choroidal vasculature and RPE cells, implicating choroidal macrophage depletion or malfunction as a factor in aging- and AMD-associated pathology.

Essential revisions:

1) Please add the number of mice and eyes assessed in each sub figure.

2) The purpose of this manuscript is to show a relationship with macrophages and visual function, if there is experimental recovery of macrophages is there recover of visual function? This significantly strengthens the conclusions.

3) Were the mice perfused before the eyes were collected. How much of the observed staining is intravascular cells? It is difficult with the whole mounts to determine if the decrease in vascular structure is because of lost functional choroidal cells, or because of reduced number of white blood cells in the blood circulation causing the vessels to look smaller and less stained.

4) None of the figures have staining controls presented. Secondary antibody staining controls should be provided, at least as a supplement.

---

## [Author Response]

Essential revisions:1) Please add the number of mice and eyes assessed in each sub figure.

We apologize for the omission. The number of mice and eyes assessed in the experiments have been added to the figure legend in all the supplementary figures.

2) The purpose of this manuscript is to show a relationship with macrophages and visual function, if there is experimental recovery of macrophages is there recover of visual function? This significantly strengthens the conclusions.

We appreciate the reviewer’s query. While we have associated the depletion of choroidal macrophages with degenerative changes in choroidal and RPE structure, as well as decreases in ERG physiological function, we found that the repopulation of choroidal macrophages was associated with the recovery of some but not all of these phenotypes. Phenotypes such as choroidal thickness were stabilized upon macrophage repopulation but did not recover back to baseline levels.

With regards to the potential recovery of ERG function with macrophage repopulation, we have now added Figure 8—figure supplement 2 to describe the changes in ERG function. This experiment involved the analysis of two separate groups of age-matched animals: (1) the continuous depletion group which was administered a PLX5622 diet continuously over 7 weeks and assessed with ERG at baseline, 3 weeks and 7 weeks, and (2) the depletion-repopulation group which was administered PLX5622 for only the first 3 weeks and resumed on a regular diet thereafter; these animals were also assessed at baseline, 3 weeks and 7 weeks. We found that in the continuous depletion group, the scotopic a- and b-wave amplitudes decreased significantly from baseline at 3 weeks of depletion. and then decreased only a small amount further at 7 weeks of depletion. In the depletion-repopulation group, a similar decrease in the scotopic a- and b-wave amplitudes were observed at 3 weeks of depletion, but the scotopic a- and b-wave amplitudes stabilized and were unchanged between 3 to 7 weeks during macrophage repopulation. These changes in ERG changes upon macrophage repopulation resembled the changes in choroidal thickness which stabilized but did not recover after macrophage repopulation.

In the Results section, we have added the following description:

“In addition to the RPE morphology and gene expression changes, the scotopic a- and b-wave amplitudes decreased significantly at 3 weeks of depletion in the continuous depletion group and decreased slightly further in the scotopic b-wave amplitude at 7 weeks of depletion while the scotopic a-wave amplitude remained stable. However, in the depletion-repopulation group, while similar decrease in the scotopic a- and b-wave amplitudes were observed at 3 weeks of depletion, the scotopic a- and b-wave amplitudes were unchanged between 3 to 7 weeks during macrophage repopulation (Figure 8—figure supplement 2A).”

“Taken together, these data demonstrate that structural and gene expression changes induced in the RPE-choroid complex upon macrophage depletion were significantly ameliorated or stabilized upon macrophage repopulation, supporting the notion of a homeostatic supportive function for choroidal macrophages.”

The following figure legend was added to Figure 8—figure supplement 2:

*“*Figure 8—figure supplement 2. Changes in ERG amplitudes during macrophage depletion and subsequent macrophage repopulation. […] n = 10 eyes from 5 animals in each treatment group.”

3) Were the mice perfused before the eyes were collected. How much of the observed staining is intravascular cells? It is difficult with the whole mounts to determine if the decrease in vascular structure is because of lost functional choroidal cells, or because of reduced number of white blood cells in the blood circulation causing the vessels to look smaller and less stained.

All experimental mice referenced in this section were perfused prior to enucleation with DiI-containing perfusate; the incorporation of DI into endothelial cell membranes enabled labeling of the choroidal vascular structure. It is possible that the removal of intravascular cells may have introduced some alterations in the vascular structure of choroidal vessels, however because all animals were similarly perfused, the detected vascular structural differences between depleted and non-depleted animals do not arise from this potential influence. These differences between the DiI-labelled vascular structures of depleted vs. non-depleted animals were corroborated by comparisons using H&E-stained choroidal sections and in vivo OCT imaging – techniques that do not involve vascular perfusion. As such, we are of the opinion that vascular perfusion did not constitute a confounding factor in our study.

For clarification, we added the sentence “to visualize the choriocapillaris, vessels were perfused and labeled with Dil lipophilic dye”in the Results section.

4) None of the figures have staining controls presented. Secondary antibody staining controls should be provided, at least as a supplement.

We had used the same secondary antibody in the immunohistochemical analysis for VEGF, PEDF and TGFβ1 in the vibratome sections. Image panels showing secondary antibody staining controls for VEGF, PEDF and TGFβ1 in vibratome sections are now added as the Figure 5—figure supplement 2; these showed no positive staining in the absence of the primary antibody.

The following figure legend was added to Supplementary file 3. “Figure 5—figure supplement 2. Secondary antibody staining control. […] n = 3 animals in each treatment group.”